# How the visual brain can learn to parse images using a multiscale, incremental grouping process

**Sami Mollard**[1]*, **Sander M. Bohte**[2,3], **Pieter R. Roelfsema**[1,4,5,6]

**1** Department of Vision & Cognition, Netherlands Institute for Neuroscience, Amsterdam, The Netherlands, **2** Machine Learning Group, Centrum Wiskunde & Informatica, Amsterdam, The Netherlands, **3** Swammerdam Institute for Life Sciences, University of Amsterdam, Amsterdam, Netherlands, **4** Laboratory of Visual Brain Therapy, Sorbonne Université, Institut National de la Santé et de la Recherche Médicale, Centre National de la Recherche Scientifique, Institut de la Vision, Paris, France, **5** Department of Integrative Neurophysiology, Center for Neurogenomics and Cognitive Research, VU University, Amsterdam, The Netherlands, **6** Department of Neurosurgery, Amsterdam University Medical Center, Amsterdam, The Netherlands

* s.mollard@nin.knaw.nl (SM); p.roelfsma@nin.knaw.nl (PRR)

## Abstract

Natural scenes usually contain many objects that need to be segregated from each other and the background. Object-based attention is the process that groups image fragments belonging to the same objects. Curve-tracing tasks provide a special case, testing our ability to group image elements of an elongated curve. In the brain, curve-tracing is associated with the gradual spread of enhanced neuronal activity over the representation of the traced curve. Previous studies demonstrated that the tracing speed is higher if curves are far apart than if they are nearby. One hypothesis is that a larger distance between curves permits activity propagation in higher visual cortical areas. In these higher areas receptive fields are larger and connections exist between neurons representing image regions that are farther apart (Pooresmaeili et al., 2014). We propose a recurrent architecture for the scale-invariant tracing of curves and objects. The architecture is composed of a feedforward pathway that dynamically selects the appropriate scale for tracing, and a recurrent pathway for propagating enhanced neuronal activity through horizontal and feedback connections, enabled by a disinhibitory loop involving VIP and SOM interneurons. We trained the network using a biologically plausible reinforcement learning scheme and observed that training on short curves allowed the networks to generalize to longer curves and 2D-objects. The network chose the scale based on the distance between curves and the width of objects, just as in human psychophysics and the visual cortex of monkeys. The results provide a mechanistic account of the learning and execution of multiscale perceptual grouping in the brain.

**Data availability statement:** All the code used to train the networks and to analyze the data is available on the following GitHub address: https://github.com/samimol/multiscale_tracing.

**Funding:** This research has received funding from the European Union's Horizon 2020 Framework Programme for Research and Innovation under the Specific Grant Agreement No. 945539 (Human Brain Project SGA3, Task 3.7, P.R.R, S.M.B.), Horizon Europe (ERC advanced grant 101052963 "NUMEROUS", P.R.R.), NWO (Crossover grant 17619 "INTENSE", P.R.R. and NWO- OCENW. KLEIN.178, S.M.B.), "DBI2", a Gravitation program of the Dutch Ministry of Science (S.M.B.), and Agence Nationale de la Recherche (AN) within Programme d'investissement d'avenir, Institut Hospital Universitaire FORESIGHT (ANR-18-590 IAHU-0001, P.R.R.). We acknowledge the use of Fenix Infrastructure resources, which are partially funded from the European Union's Horizon 2020 research and innovation programme through the ICEI project under the grant agreement No. 800858. The funders had no role in study design, data collection and analysis, decision to publish, or preparation of the manuscript.

**Competing interests:** The authors have declared that no competing interests exist.

## Author summary

In our perception, image elements that belong to the same object are grouped by object-based attention. Object-based attention corresponds to an enhanced neuronal representation of the image elements that are grouped in perception, in multiple areas of the visual cortex. During perceptual grouping tasks, this enhanced neuronal activity spreads gradually over an object representation, with a speed that depends on the distance between the relevant object and other objects. Here we propose a neuronal mechanism that learns the scale-invariant spread of object-based attention and accounts for psychophysical observations in human observers and the pattern of neuronal activity in the visual cortex of monkeys. This work sheds light on the mechanisms for multiscale object-based attention in the visual cortex.

## Introduction

Natural scenes usually contain several objects that need to be segregated from each other and from the background to guide behavior. Previous studies demonstrated that neurons in areas of the visual cortex that encode the elements of a relevant object incrementally enhance their firing rate during perceptual grouping tasks [1]. This neuronal process corresponds to the gradual spread of "object-based attention" in perceptual psychology [2,3]. The rules determining what groups with what were already studied in the first half of the 20th century by the Gestalt psychologists [4,5]. One of their rules is connectedness, because image elements that are connected to each other tend to belong to the same object and group in our perception. Another rule is that of good continuation, describing that collinear contours usually belong to the same object.

  Early theories of perception suggested that Gestalt rules are applied pre-attentively and in parallel across the visual field [6], and indeed, some basic forms of grouping related to, for example, the perception of shape take place in parallel across the visual field [2,3]. However, grouping is flexible, because we can also group the features of objects that we never saw before. These groupings appear to form incrementally if image elements are grouped indirectly, via other image elements. On example of such seriality occurs in curve tracing tasks, in which participants report if two elements belong to the same curve (Fig 1A). Jolicoeur et al. [7] showed that the reaction time (RT) is proportional to the length of the curve that subjects have to trace. This serial process is also reflected by neurons in the visual cortex. Neurons in the visual cortex of monkeys [1] and humans [8] with a receptive field (RF) on the target curve increase their firing rate relative to neurons with RFs on a distractor curve (Fig 1A,1B). This response enhancement does not occur during the initial feedforward visual response, triggered by the onset of a visual stimulus in the RF, but after a delay. The latency of the response enhancement depends on the distance between the start of the tracing process and the position of the RF. It occurs later for neurons

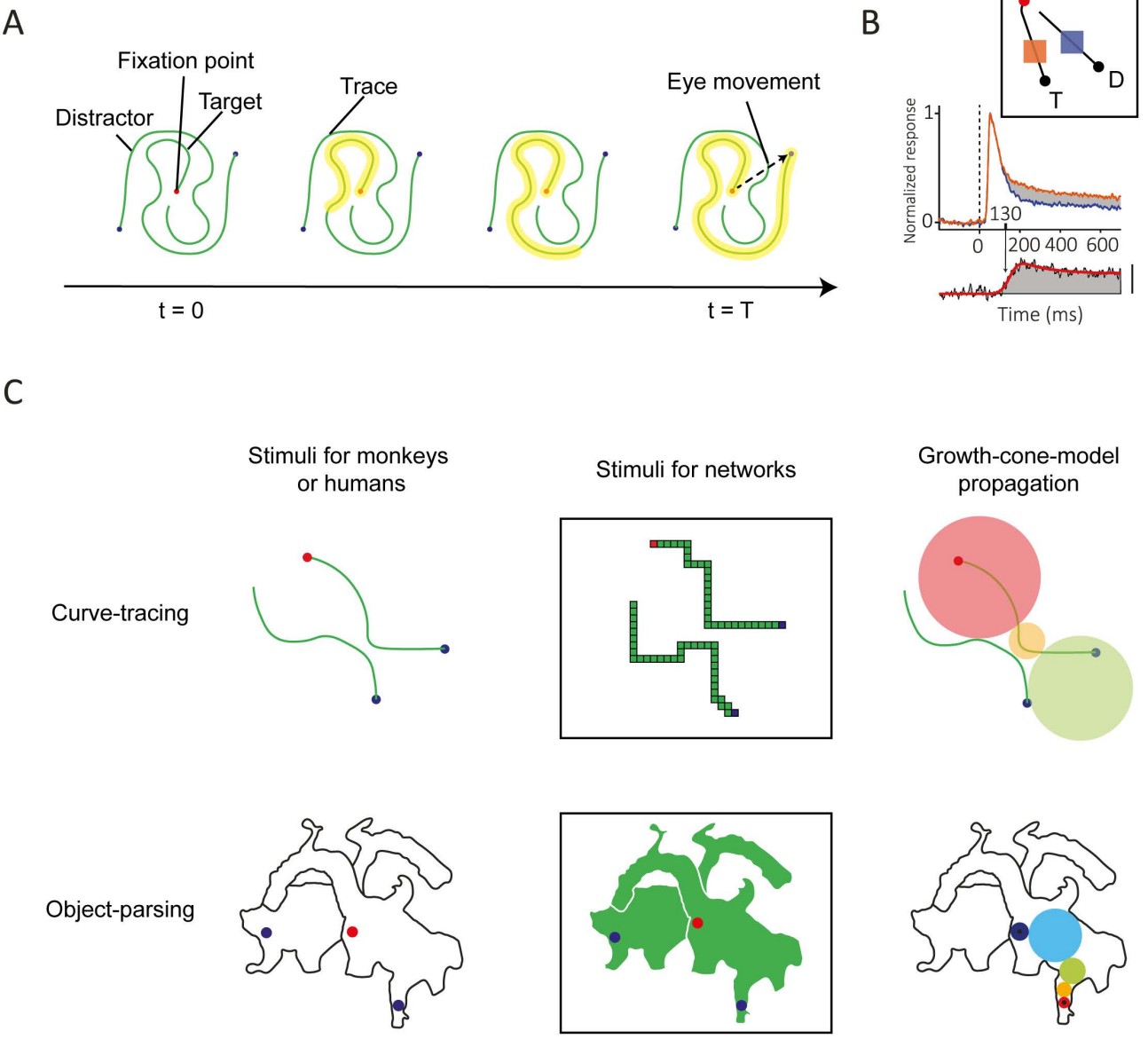

**Fig 1. Tasks used to probe the dynamics of object-based attention. A.** In this example curve tracing task, the subject must make an eye movement toward the blue dot connected to the red fixation point. Extra neuronal activity gradually spreads across the representation of the target curve in the visual cortex (yellow). **B.** Activity of neurons in the primary visual cortex of monkeys elicited by a target curve (orange) or on a distractor curve (blue). Inset, curve-tracing stimulus in which the V1 RF either fell on the target (orange) or distractor curve (blue). The panel below shows the time-course of attentional modulation, which is the difference in V1 activity between conditions. The target curve is labelled with enhanced activity (adapted from [20]). **C.** Curve tracing and object grouping. In the curve tracing task, a growth-cone model of attention can account for the time-course of neuronal responses by proposing that enhanced activity spreads at multiple scales. If curves are far apart, the model uses large RFs (large circles) and smaller RFs when curves are nearby. The object grouping task is analogous to the curve-tracing task but takes place within two-dimensional objects. Subjects determine which dots are on the same image component.

with RFs farther along the curve [9], in accordance with the gradual spread of enhanced activity over the representation of the target curve (Fig 1A). This spread of an enhanced response corresponds to the gradual spread of object-based attention across the relevant curve [10,11].

The curve-tracing speed depends on the distance between the target curve and the distractor curves and on the curvature. Tracing slows down if distractors are nearby the target curve [9,12] or if it has a high curvature [13]. Pooresmaeili et al. [9] tested the influence of the distance between RFs on tracing speed in the visual cortex of monkeys. In their experiment there was a bottleneck where the target curve came close to the distractor (Fig 1C, top). Narrowing of the gap between the curves delayed the onset of the response enhancement, but only for parts of the target curve that were behind the bottleneck. A possible explanation of the influence of distance between curves is that the spread of enhanced neuronal activity across the target curve occurs in multiple cortical areas with different RF sizes [9]. When the target curve is close to a distractor, the propagation of the enhanced activity would occur in low-level areas, such as the primary visual cortex, where RFs are sufficiently small to not fall on both curves. In these low-level areas, connections between neighboring neurons interconnect neurons with nearby RFs so that tracing speed is low. When the curves are farther apart, the spreading could occur in higher areas where the neurons have larger RFs and connections bridge across larger distances in the visual field. A "growth-cone" model for the spread of object-based attention proposed that fastest progress is made in the area where RFs falling on the target curve almost touch the distractor curve (Fig 1C). An interesting implication of the growth cone model is that the RT in curve-tracing depends only little on the viewing distance from a display. When the subject approaches the stimulus, the length of the curves, measured in degrees of visual angle, increases. However, the distance between curves also increases and grouping can rely on neurons in higher visual areas with larger RFs. These two effects cancel each other and the RT remains the same [14].

Jeurissen et al. [15] extended the growth cone model to 2-D images. In their study, human participants reported whether two dots were on the same object or not. The pattern of RTs was best explained by a version of the growth-cone model in which image regions are incrementally labeled with enhanced activity. The enhanced activity started at one of the dots and propagated across the target surface through neurons with RFs with sizes that were small enough to stay within the object boundaries (Fig 1C, bottom).

In a previous study, Marić & Domijan [16] proposed an instantiation of the growth-cone model for curve-tracing in a neural network with multiple scales. Their neural network architecture used hard-coded, multi-scale Gabor filters and the synaptic weights were determined manually to select the appropriate scale. Their network qualitatively reproduced the results observed in monkey visual cortex, but the model did not yet generalize to 2-D images.

Here, we focus on neural network architectures that can learn to incrementally group elongated curves and spatially extended image regions and we test how the networks generalize across tasks. Specifically, we examine how recurrent neural networks with units with several RF sizes learn to trace curves and to fill in objects in a scale-invariant manner. Building on prior work [16,17] we developed an architecture with segregated populations of units that either represent the stimulus veridically because of pure feedforward connectivity or can be modulated through horizontal and feedback connections. A key innovation is the inclusion of a trainable, biologically inspired disinhibitory loop that enables the propagation of attentional modulation across a curve or across an image region [18].

Furthermore, we implement a biologically plausible learning rule that can train networks by trial-and-error. The only feedback that the model receives is a reward if it makes the correct choice, like how monkeys are trained on a curve-tracing and region-filling tasks. Specifically, we used RELEARNN [17,19] to determine how synapses modify their connection strength, using information both local in space and time.

Our approach addresses (1) how the networks learn to group image elements of the same objects, (2) the mechanisms allowing networks to learn to propagate enhanced activity at multiple spatial scales. We report that training on short curves allowed the model to generalize to long curves and to 2-D shapes. Furthermore, the networks predicted human RTs during image parsing tasks and the spread of neuronal activity in the visual cortex of monkeys.

 

## Model and task

We trained the networks on a curve-tracing task that has been used during electrophysiological recordings in the visual cortex of monkeys [9] (Fig 1A,1B). The monkeys first had to direct their gaze to the fixation point before the stimulus appeared, but in the version of the task for the network, the entire stimulus was presented at once. We also tested how well models that are proficient in curve-tracing generalize to the parsing of 2-D image regions [15,21]. The task of the model was to select a blue pixel on a curve or object that was cued with a red pixel (Fig 1C) as target for an eye movement.

As in previous work [17], we included a feedforward and recurrent processing group of units (Fig 2A). Neurons in the feedforward group only receive feedforward input and propagate the information to higher layers. They represent the stimulus veridically and are responsible for the selection of the appropriate scale. Neurons in the recurrent group receive feedforward connections from lower layers, feedback from higher layers and horizontal connections from neighboring units in the same layer. They are responsible for the propagation of enhanced neuronal activity. The presence of some units that propagate the enhanced activity and others that do not is in accordance with neurophysiological results [22–25]. Neurons that engage in incremental grouping are enriched in layers 2, 3 and 5 of the visual cortex, whereas neurons that veridically represent the feedforward input are mostly situated in layers 4 and 6 [22–24]. In the model, units in the feedforward group gated the units of the recurrent group, so that they could not spread enhanced activity if the corresponding feedforward unit was not active (Fig 2A). This feature prevented the spreading of enhanced activity into image regions not occupied by curves or objects.

## Feedforward network

The neural network had four spatial scales, and the feedforward units were responsible for scale selection. They were trained to only activate if the stimulus in their RF unambiguously belonged to a single object (see below) to prevent "spilling" of enhanced activity from the target to the distractor object in the recurrent network. In the curve-tracing task, feedforward units detected whether all pixels in their RF were connected to each other and colinear (Fig 2B). Hence, the model needed to select a finer spatial scale for propagation if the target and distractor curves fell in the same RF, or if the target curve had a high curvature [13]. In the object-parsing task, the stimulus inside a receptive field is unambiguous if it doesn't contain a boundary (Fig 2B). Upon presentation of the stimulus, feedforward units propagated activity to higher layers if the stimulus in their RF is unambiguous, resulting in a multiscale base-representation (Fig 2C).

The activity of feedforward units was computed as follows: first, an input image with three feature channels (R, G, B) was projected onto a single feature map of the same spatial dimensions (Fig 2D):

$$X^0 = ReLU\left(FF^0 \otimes \boldsymbol{X}\right)$$

(1)

Where $\boldsymbol{X} \in \mathbb{R}^{H \times W \times 3}$ is the input image and $FF^0 \in \mathbb{R}^{1 \times 1 \times 3 \times 1}$ is a 1x1 kernel with a stride of 1 and $\otimes$ indicates convolution. This projection $X^0$ served as a shared representation and forms the input for all layers and scales. There was a direct feedforward projection from the feature map units to each scale, so that the activity in, e.g., layer 4 did not need to pass through layers 1, 2 and 3 (Fig 2D).

The activity of feedforward units at all scales $l$ was computed as follows: $X^0$ was first passed through a convolutional layer with 20 feature maps, receptive field size $K$, and stride 1 to compute an intermediate feedforward activity in each layer $X^l_{int}$:

$$X^l_{int} = ReLU\left(FF^l_{int} \otimes X^0\right)$$

(2)

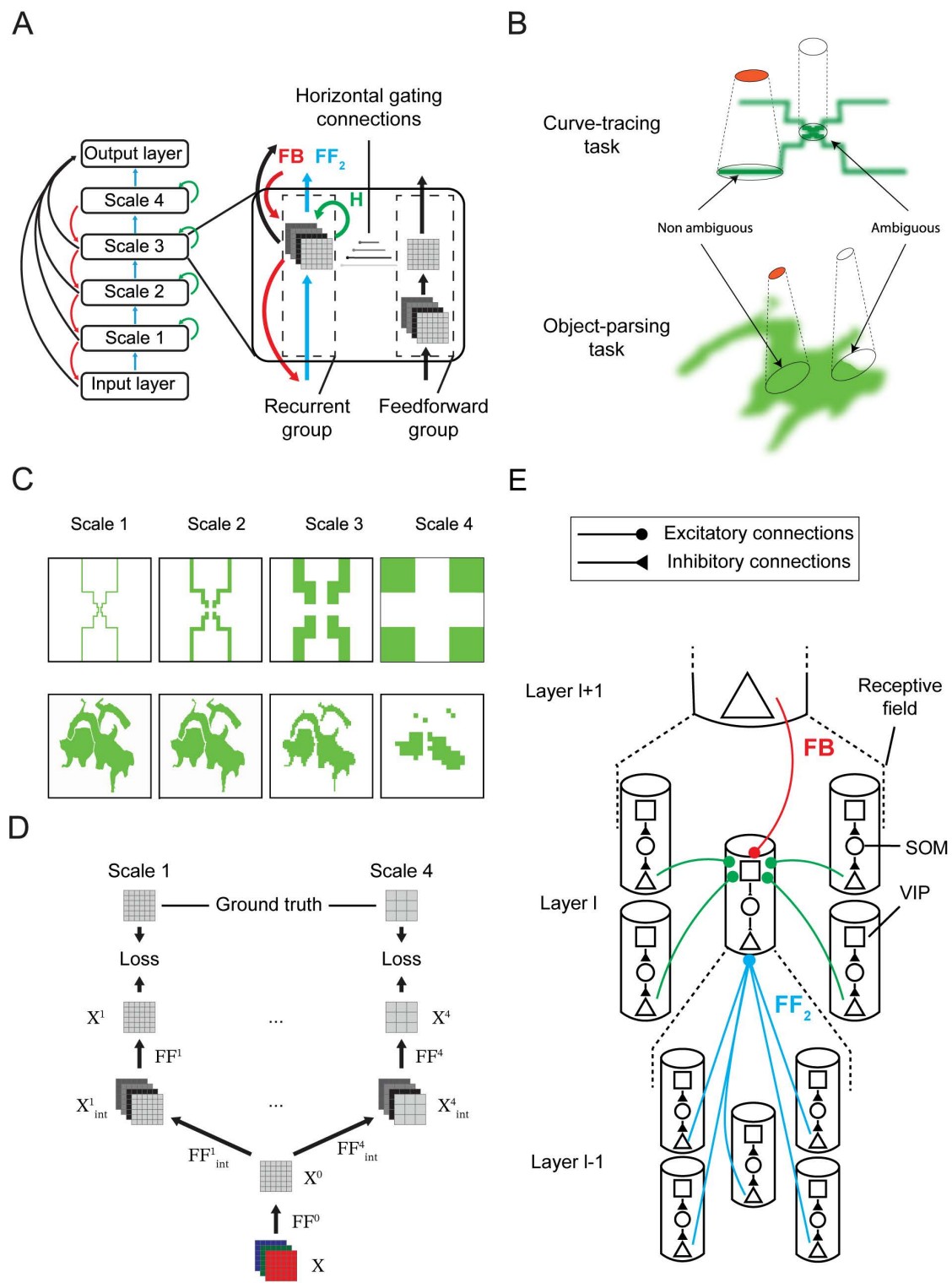

**Fig 2. Model. A.** The network incorporated four spatial scales. Units in the feedforward network are responsible for scale selection, whereas units in the recurrent network spread enhanced activity. This spread is gated by feedforward units with overlapping RFs. **B.** Scale selection in the feedforward network. In the curve tracing task, units in the feedforward group are only active (orange) if the image elements in the RF are connected and colinear. The cells are not active otherwise so that the enhanced activity cannot spill from the target to the distractor curve. In the object-parsing task, feedforward units are active if their RF falls on a homogeneous image region. **C.** Activity in the feedforward network in the curve tracing (top) and the object parsing

task (bottom). The representation in higher layers is coarser, but RFs are larger and propagation can proceed faster. **D.** Architecture and training procedure of the feedforward group. The input image is first passed through the convolutions **FF**$^0$ to produce a representation **X**$^0$ shared across all scales. Each scale is trained separately with its own ground truth and loss (only two scales are illustrated). **E.** Disinhibitory interactions between recurrent units. Pyramidal neurons receive feedforward from pyramidal units in the layer below. They also receive top-down input from units in the layer above and horizontal input from their neighbors through a disinhibitory loop composed of VIP and SOM interneurons.

Where $FF^l_{int} \in \mathbb{R}^{K \times K \times 1 \times 20}$ with $K = 3^{L-1}$ for $L \in \{1, 2, 3, 4\}$ and $X^l_{int} \in \mathbb{R}^{H \times W \times 20}$ is the intermediate representation. The final feedforward output for layer $l$, used to gate the dynamics of the recurrent network, was computed as:

$$X^l = ReLU(\sigma\left(FF^l \otimes X^l_{int}\right) - 0.7)$$

(3)

where $FF^l \in \mathbb{R}^{K \times K \times 20 \times 1}$ with $K = 3^{L-1}$ for $L \in \{1, 2, 3, 4\}$, applied with a stride $K$, $X^l \in \mathbb{R}^{H \times W \times 1}$ modulated the recurrent state and $\sigma$ is a sigmoidal non-linearity. Hence, each scale was associated with an independent two-layer neural network with shared input $X^0$ (Fig 2D). The resulting activity $X^l$ was used to gate the recurrent dynamics of the main network.

## Recurrent network

Units of the recurrent network learned to spread enhanced neuronal activity over the representation of the target object, to label it as one coherent perceptual group. They were connected by feedforward, horizontal and top-down connections. The propagation of enhanced neuronal activity relied on disinhibition, which was advantageous because we used the RELEARNN learning rule (see below) [19] requiring the network to reach a stable state, which is not guaranteed with ReLU nonlinearities (Material and Methods). The disinhibitory connection scheme (Fig 2E) was inspired by the interactions between inhibitory neurons in the mouse visual cortex during figure-ground segregation [18] and has been used in previous models of attentional selection [26,27]. It ensured stable and expressive networks that learned to trace curves with an arbitrary length.

Specifically, the feedback and horizontal connections activated vasoactive intestinal peptide-expressing (VIP) interneurons. The VIP neurons inhibit somatostatin-expressing (SOM) interneurons, which inhibit pyramidal neurons [18,28]. Hence, VIP neurons disinhibited the pyramidal neurons. During curve-tracing, VIP neurons incrementally disinhibited pyramidal units that represent the target object, which is thereby labeled with enhanced activity. The advantage of the disinhibitory scheme is that the activity of pyramidal neurons cannot be higher than what it would be without SOM inhibition, preventing run-away excitation that can occur in recurrent networks with excitatory units that directly excite each other.

We will now describe the recurrent network with an input layer ($l = 0$) and multiple hidden layers ($l \geq 1$), each containing three interacting populations: pyramidal units $Y^l$, VIP interneurons, and SOM interneurons. The network activity evolves over discrete time steps t.

**Input layer ($l = 0$).** The input layer receives the stationary sensory stimulus $X$. Its activity is modulated by feedback from the first hidden layer and by lateral inhibition. It acted as a "blackboard" [29] between the raw sensory input and the higher layer representations, enriched by modulatory feedback signals and able to propagate this information forward again.

VIP neurons in the input layer are driven exclusively by feedback from the first hidden layer:

$$VIP^0(t) = \Theta\left(FB^0 \otimes Y^1(t-1)\right)$$

(4)

with feedback weights

$$FB^0 \in \mathbb{R}^{C_0 \times C_1 \times K \times K}, \ K = 3$$

with stride 1. Here $C_0$ is the number of channels in the input layer, $C_1$ is the number of channels in the first hidden layer, and $\Theta$ is a clipped linear function defined by

$$\Theta(x) = \begin{cases} x, x < 1 \\ 1, otherwise \end{cases} \tag{5}$$

SOM activity is defined as:

$$SOM^0(t) = ReLU\left(\mathbf{1} - VIP^0(t)\right), \tag{6}$$

where $\mathbf{1}$ denotes a tensor of ones with matching dimensions.

Pyramidal activity in the input layer is not directly driven by feedforward input (unlike in the hidden layers). The units receive top-down feedback from the first hidden layer, which is filtered through lateral inhibition. The feedback-modulated inhibitory signal is then gated by the sensory input image, allowing feedback to selectively weight the color channels, while preserving the spatial structure imposed by the sensory input. The activity of pyramidal units was defined by:

$$Y_0(t) = ReLU\left(\phi\left(\mathbf{X}\right) \odot \left(-LI^0 \otimes SOM^0(t)\right)\right) \tag{7}$$

where lateral inhibitory kernel $LI^0$ represented self-connections, with a single nonzero value at the center of every $3 \times 3$ kernel. The SOM inhibition was local and it did not mix information across spatial positions. The inhibitory weights from the SOM cells to the pyramidal cells satisfied

$$LI^0 \in \mathbb{R}^{C_0 \times C_0 \times K \times K}, \quad K = 3$$

applied with stride 1. The $\phi$ is a gating function:

$$\phi(x) = \frac{|100x|}{\sqrt{1+(100x)^2}}, \tag{8}$$

ensuring that propagation of the enhanced activity can only occur if the scale-selecting feedforward units $\mathbf{X}$ are active. $\odot$ is the Hadamard (element-wise) product.

**Hidden layers ($l \geq 1$).** Hidden layers operated at progressively coarser spatial resolutions $H^l \times W^l$, where $H^l = \frac{H}{3^{l-1}}$ and $W^l = \frac{W}{3^{l-1}}$. This reduction in spatial resolution and the associated increase in RF size were inspired by the hierarchical organization of the visual cortex, but did not precisely map onto the resolution and RF sizes in specific visual cortical areas. Each hidden layer had $C^l$ feature channels ($C^l = 1$ for $l = 1$; 6 otherwise). VIP activity in the hidden layers integrated feedback and horizontal signals:

$$\mathbf{VIP}^l(t) = \Theta\left(H^l \otimes \mathbf{Y}^l(t-1) + FB^l \otimes \mathbf{Y}^{l+1}(t-1)\right), \tag{9}$$

where $FB^l \in \mathbb{R}^{C_l \times C_{l+1} \times K_l \times K_l}$ with $K_l = 3$ and a stride of 3. These feedback kernels $FB^l$ used transposed convolution, such that a neuron in layer $l$ only received feedback from neurons in layer $l+1$ with overlapping RFs, restricting feedback modulation to spatially aligned units (red in Fig 2E). Finally, horizontal kernels $H^l \in \mathbb{R}^{C_l \times C_l \times 3 \times 3}$ had a stride of 1 and were restricted to the von Neumann neighborhood that corresponds to the four nearest neighbors (up, down, left right). SOM activity was given by

$$SOM^l(t) = ReLU\left(\mathbf{1} - VIP^l(t)\right).$$

(10)

Finally, pyramidal neuron activity evolved according to:

$$Y^l(t) = ReLU\left(\phi\left(X^l\right) \odot \left(FF_2^l \otimes Y^{l-1}(t) - SOM^l(t)\right)\right)$$

(11)

with feedforward weights $FF_2^l \in \mathbb{R}^{C_l \times C_{l-1} \times K_l \times K_l}$, where $K_l$=3, for $l = 1$ and $K_l$=1 otherwise, with the stride equal to $K_l$. We note that feedforward kernels $FF_2^l$ were fully connected within a 3x3 receptive field, ensuring that each neuron in layer $l$ received feedforward input from all neurons with overlapping RFs in layer $l-1$ (cyan connections in Fig 2E). Note that $FF_2^l$ denotes the feedforward connections of the recurrent network, which differ from the feedforward connections of the scale selection feedforward network described above.

## Output layer

The output layer was retinotopically organized and units in the output layer learned to represent the Q-value [30], which is the expected reward for the selection of an eye-movement to one of the pixels. Each output unit corresponded to a pixel and integrated information across all scales via skip connections from the input layer and all hidden layers. Without these skip connections, the higher layers, which have a lower resolution, dominated the decision process, preventing the network from selecting the correct pixel for an eye movement.

The Q-value map was read out when the network dynamics converged to a stable state at time T. Let $Y^l(T)$ denote the pyramidal activity at layer $l$ and time $T$. The activity in the output layer was given by:

$$Q = \sum_{l=0}^{4} W^l \otimes Y^l(T)$$

(12)

where $W^l$ denotes the skip-connection kernel from layer $l$ to the output layer.

The skip connections from the input layer ($l = 0$) were implemented as $1 \times 1$ convolutions, preserving full spatial resolution, and skip connections from higher layers were transposed convolutions aligned with the output layer: a neuron in layer $l$ could only project to the $3^l \times 3^l$ output units representing pixels in its receptive field. We note that these skip connections did not suffice for solving the task for stimuli extending beyond two receptive fields at the largest scale.

During training, the model chose to make an eye-movement toward the position with the highest Q-value with probability $1 - \varepsilon$. With probability $\varepsilon$ other actions were explored, by sampling a random action from the Boltzmann distribution $P_B$:

$$P_B(a) = \frac{\exp(Q_a)}{\sum_k \exp(Q_k)}$$

(13)

We simulated the selection of an eye movement but not the eye-movement itself or the shift of the visual image caused by it.

## Training

We trained the networks in two phases. We first trained the feedforward network and we trained the recurrent network thereafter. The curve-tracing stimuli were random curves on a 36x36 grid, and stimuli for the object-parsing task were random objects on a 72x72 grid.

**Training of feedforward networks.** We trained different feedforward networks for the curve-tracing and object-parsing tasks. Feedforward units for the curve-tracing task detected whether all image elements inside their RF were colinear and connected (using 50,000 stimuli for training) whereas units for the object-parsing task detected whether all inputs in their RF were active (10.000 stimuli) (Fig 2B, 2C).

We trained each spatial scale separately with backpropagation during 80 epochs using the Adam optimizer [31] and a learning rate of $10^{-3}$. Upon presentation of an input image, a cross-entropy loss was computed independently at each spatial scale (Fig 2D) and the training objective was based on summed losses across scales. Because each scale has its own loss term, the scale-specific weights updates only depended on the error signals at that scale. The only exception concerns the weights $FF^0$ which are shared across scales and therefore receive gradient contributions from all scale-specific losses (Fig 2D).

We verified the accuracy with 100 stimuli for each task. Although we used error-backpropagation, we note that it could be replaced by a biologically plausible reinforcement learning rule [32]. We assumed that the tuning of feedforward units had emerged during visual experience [33–37], prior to training on curve-tracing or object-parsing. We froze the weights of the feedforward network during the subsequent recurrent network training phase. We note that the feedforward architecture processes the various spatial scales in parallel, and its units only combine information within a limited spatial neighborhood so that they are unable to group pixels outside this range.

**Training of the recurrent network.** Weights in the recurrent network were updated using RELEARNN, a local learning rule inspired by the Almeida-Pineda algorithm [2,17,19,38] that uses three phases. In the first phase, neural activity propagated through the network until convergence to a stable state. We considered that a stable state was reached if the activity of neurons between two consecutive timesteps was the same or after 30 timesteps. In the second phase, the network selected an eye movement, as described above, and an attention signal originating from the winning action propagated through an accessory network. This attentional feedback signal is proportional to the influence of each unit on the selected action. This accessory network can be conceived of as a linearized, transposed version of the activity propagation network and can therefore assign credit to synapses (for details see ref. [17]). In the third phase, the network received a reward $r$ of 1 unit in case the eye movement was correct and 0 otherwise. It then computed a reward prediction error $\delta$:

$$\delta = r - Q_a, \tag{14}$$

where $Q_a$ is the activity of the selected output unit. The reward-prediction error $\delta$ could be broadcasted to the whole network by a neuromodulatory signal [39]. The reward-prediction error $\delta$ and the attentional feedback signal are available at all synapses and determine the weight update, together with the pre- and post-synaptic activity. Hence, the required information is available locally at the synapse, making RELEARNN biologically plausible. We used a curriculum for training [17], a strategy also used to train monkeys. The network was first presented with curves of 3 pixels. Once the network achieved 85% accuracy during a test phase with fixed weights and no exploration, we added one pixel to the curve and repeated this procedure until the curves were 7 pixels long.

To reduce the computation time, we used weight sharing, although this is biologically implausible. In previous work, we showed that networks with a similar structure learned to trace curves with or without weight-sharing, but that the number of trials needed to learn the task without weight-sharing was ~7 times larger [17]. Hence, our results are likely to generalize to learning rules without weight sharing.

## Results

### Curve-tracing task

We trained 5 networks on a curve-tracing task, which was a variant of a task that has been used for electrophysiological recordings in the visual cortex of monkeys [9] (Fig 1A,1C). In the version of the task used here, the stimulus (illustrated in

Fig 1C) was presented at once (Fig 3A), and the network was rewarded if it selected an eye movement to a blue pixel that was connected by a curve to a red pixel. We trained the networks with a curriculum in which the length of the curves was increased until they were 7 pixels long. We used a criterion of 85% accuracy to determine that a network had converged, but all networks reached an accuracy of 100% on the curve-tracing task, within an average of 23,200 trials. To examine whether the networks memorized specific curve configurations [40] or learned a general grouping rule we generated fifteen curves with a length of 30 pixels. The accuracy of the networks was 100%, confirming that they learned a general solution. Units of the recurrent network had learned to iteratively spread enhanced activity across the representation of the target curve.

The model learned to exploit the disinhibitory circuit to propagate enhanced activity through the recurrent network, starting from the red pixel (Fig 3A). In the baseline state, SOM interneurons are spontaneously active and suppress pyramidal neurons with overlapping RFs. Feedforward input alone is insufficient to overcome this inhibition. Through trial-and-error learning, the network increased the weights $FF_2^1$ from input units coding for red to the pyramidal neuron in the first hidden layer, which now overcame SOM-mediated inhibition ($T = 1$ in Fig 3A). This activation recruited adjacent VIP interneurons via horizontal and feedback connections. The VIP neurons, in turn, inhibited the SOM interneurons, thereby disinhibiting pyramidal cells representing the adjacent pixels of the target curve. Repetition of this disinhibitory interaction led to a stepwise propagation of enhanced activity along the target curve. We compared the present disinhibitory circuit to our previous excitatory recurrent model in ref. [17] (Fig A in S1 Text). The previous excitatory model exhibited progressive attenuation of the response enhancement along longer curves, resulting in impaired generalization to new and longer stimuli. In contrast, the present disinhibitory network maintained a stable activity difference between target and distractor representations, enabling reliable generalization to longer curves.

We next examined how the model's tracing speed depends on the distance between the target and distractor curves. We presented stimuli with a bottleneck where the target curve came close to the distractor [9] (Fig 3B) and examined the activity of units with RFs before and after the bottleneck. The latency of the response enhancement, measured as the time-step where activity reached 90% of its maximum, was shorter for units with RFs before the bottleneck than for units with RFs behind (Fig 3C). Narrowing the gap between the curves delayed the onset of the response enhancement, but only for RFs after the bottleneck (Fig 3D), just as has been observed in the visual cortex of monkeys [9] (Fig 3D). The five models that we trained learned similar strategies and there was hardly any variability in their dynamics (absence of error bars in Fig 3D).

We next examined the scales that were selected by the model (Fig 3B) and noticed that the small gap enforced the spread of enhanced activity at lower network levels, where RFs were smaller. The propagation speed was also influenced by curvature because the network resorted to spreading activity at lower network levels if image elements in the larger RFs were not colinear. This results is in agreement with experimental evidence in humans that a higher curvature decreases tracing speed [13].

## Object-parsing task

We next probed the ability of the networks to group image elements of 2-D objects, using a variant of an object-parsing task used in human participants [15,21,41]. The participants reported whether a cue fell on the same object as the fixation point. The version for the network was like the curve tracing task of the previous section. The fixation point was a red pixel, and we placed a blue pixel on the same 2-D object and another one on a second object. The network had to plan an eye movement to the blue pixel on the same object as the fixation point (Fig 4A).

The models had the same architecture as those for the curve-tracing task (Fig 2). We started with the recurrent units that been trained on the curve-tracing task and retrained their weights for the object-parsing task using the same RL process. However, we now used a feedforward network that was specifically trained for scale selection in the object-parsing task. All 5 networks reached an accuracy higher than 85% within 100 trials when tested in the object-parsing task. This

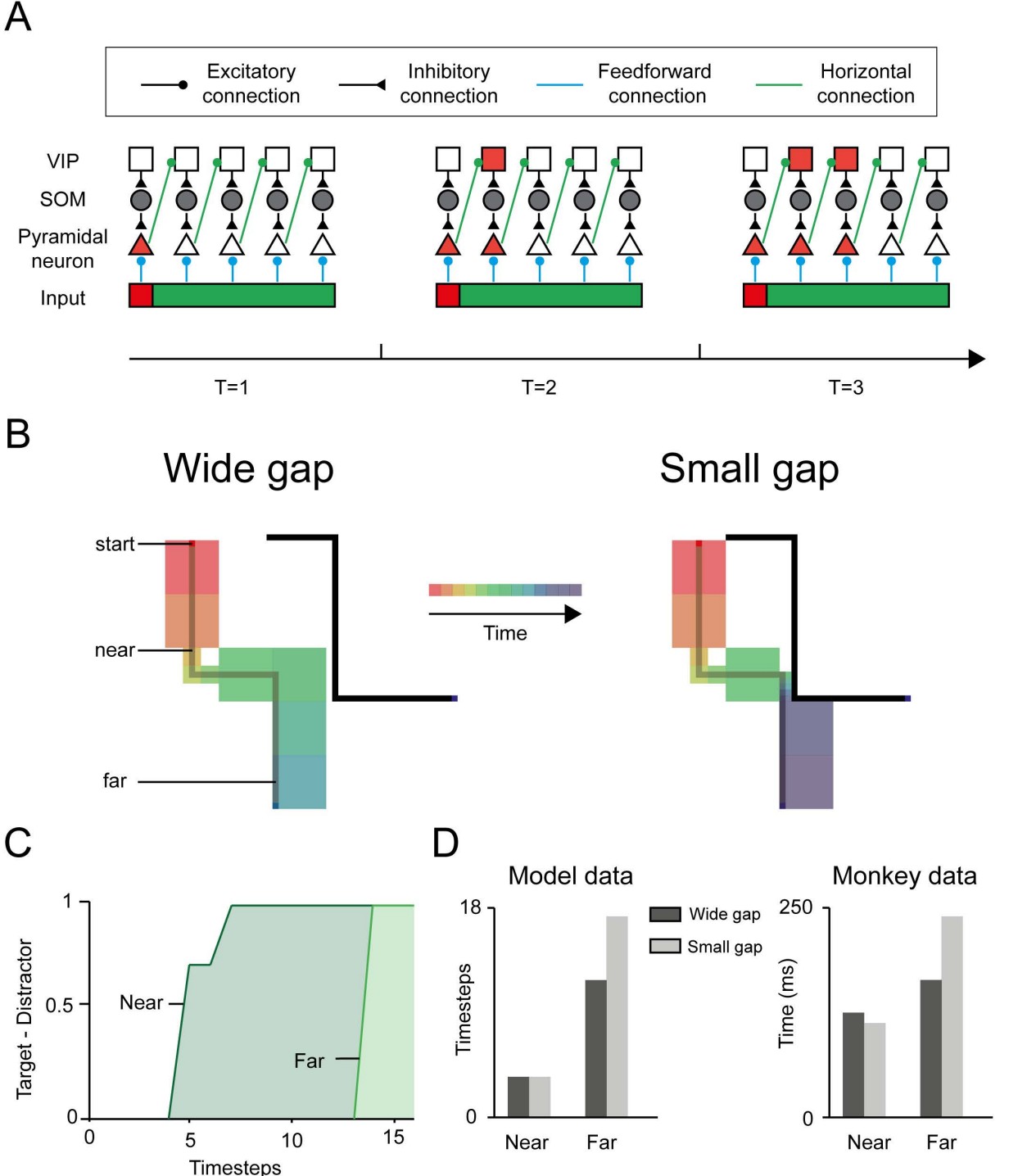

**Fig 3. Networks propagate the response enhancement at multiple scales. A.** Spreading of activity by disinhibition. At $T=1$ the visual stimulus is presented, and the red pixel activates a pyramidal unit, which in turn activates VIP units in an adjacent column through horizontal connections (green), which cancels SOM inhibition and activates the next pyramidal cell at $T=2$. The disinhibition then propagates further along the target curve. **B.** Stimuli with gaps of two sizes between the target and distractor curve (not used in training). The size of the squares represents the scale that was used by the networks and color represents the time at which the attentional tag reached the RF. When the curves are far apart, tracing occurs faster and at a larger scale. **C.** We measured the activity of units with RFs before or after the gap. The activity elicited by the target curve was higher, and the response

enhancement occurred later for units with RFs behind the gap (far) than for units with RFs before the gap (near). **D.** Latency of the response enhancement in the model (left) and in area V1 of monkeys (right). The response enhancement did not depend on gap size for near RFs before the gap. A narrow gap delayed the response enhancement of units with RFs beyond the gap.

additional training did not influence accuracy in the curve-tracing task, which remained 100%, implying that the recurrent architecture can account for both curve-tracing and object-parsing. The networks parsed the object that was cued by the red fixation point by spreading enhanced activity over its representation (Fig 4B), just as has been observed in V1 of humans with fMRI [42].

We next investigated the dynamics of the parsing process by comparing the processing time in the model to human RTs in two studies. The first experiment was by Jeurissen et al. [15] who presented scrambled shapes, which could not be recognized by the participants (Fig 4A). To model the RT, we estimated the timestep at which units in the output layer whose receptive field fell on the blue pixel enhanced their response by 90%. The model explained 55% of the variance of human RTs. The heuristic growth-cone model (Fig 1) accounted for 63% of the variance, which is close to the noise ceiling of 67%. A key difference between the growth-cone model and its implementation as neural network is the number of scales. The neural network used four scales, whereas the number of scales used by the growth-cone model was not bounded. To further examine the dependence on the number of scales, we tested models with fewer scales. We observed that the models with 2 scales and 3 scales explained significantly less variance than the model with 4 scales ($p < 10^{-3}$ and $p = 0.03$, respectively, see Methods) (Fig 4C).

We replicated these findings modeling data from a second study that presented naturalistic images from the COCO dataset [43]. We simplified the task by presenting object masks to the neural network and we measured the number of timesteps before the enhanced response reached the blue pixel on the same object as the fixation point (Fig 4A). The neural network accounted for 15% of the variance in the human RTs, which is close to the 16% accounted for by the growth cone model and to the relatively low noise ceiling in this data set of 24%. These results imply that that the neural network accounted well for the patterns of human RTs in image parsing tasks.

## Discussion

In this study, we used neural networks to understand the propagation of object-based attention in curve-tracing and object-parsing tasks and how it can be learned [44]. The processing time in curve tracing tasks increases linearly with the length of the curve that needs to be traced. In the visual cortex of monkeys [1,9] and humans [8,42], curve-tracing is associated with the gradual propagation of enhanced neuronal activity along the representation of the target curve [8]. The speed of the tracing process depends on the curvature of the target curve and on its distance to distractor curves [9,13]. Recent studies started to examine the neuronal processes responsible for the parsing of 2-D surfaces and natural objects where the processing time also depends on the distance between the image locations that need to be grouped, and on the width of the image regions connecting them [15,21].

We trained a recurrent neural network to perform a curve-tracing task using trial-and-error learning, mimicking how monkeys are trained. With minimal additional training, the same networks could also parse spatially extended objects. Remarkably, the model accounted for many neurophysiological and psychophysical findings. We reproduced the pattern of RTs in humans in curve-tracing and image-parsing tasks, including the influence of the curvature and distance between curves and the width of image regions. Furthermore, the model developed a strategy to incrementally label the relevant curve or image region with enhanced neuronal activity, just as is observed in the visual cortex. The model thereby provides a neural network implementation of the conceptual "growth-cone" model (Fig 1C) [9,13,15], explaining why grouping speed depends on the distance between curves and the width of object regions. With only 4 scales, the network's

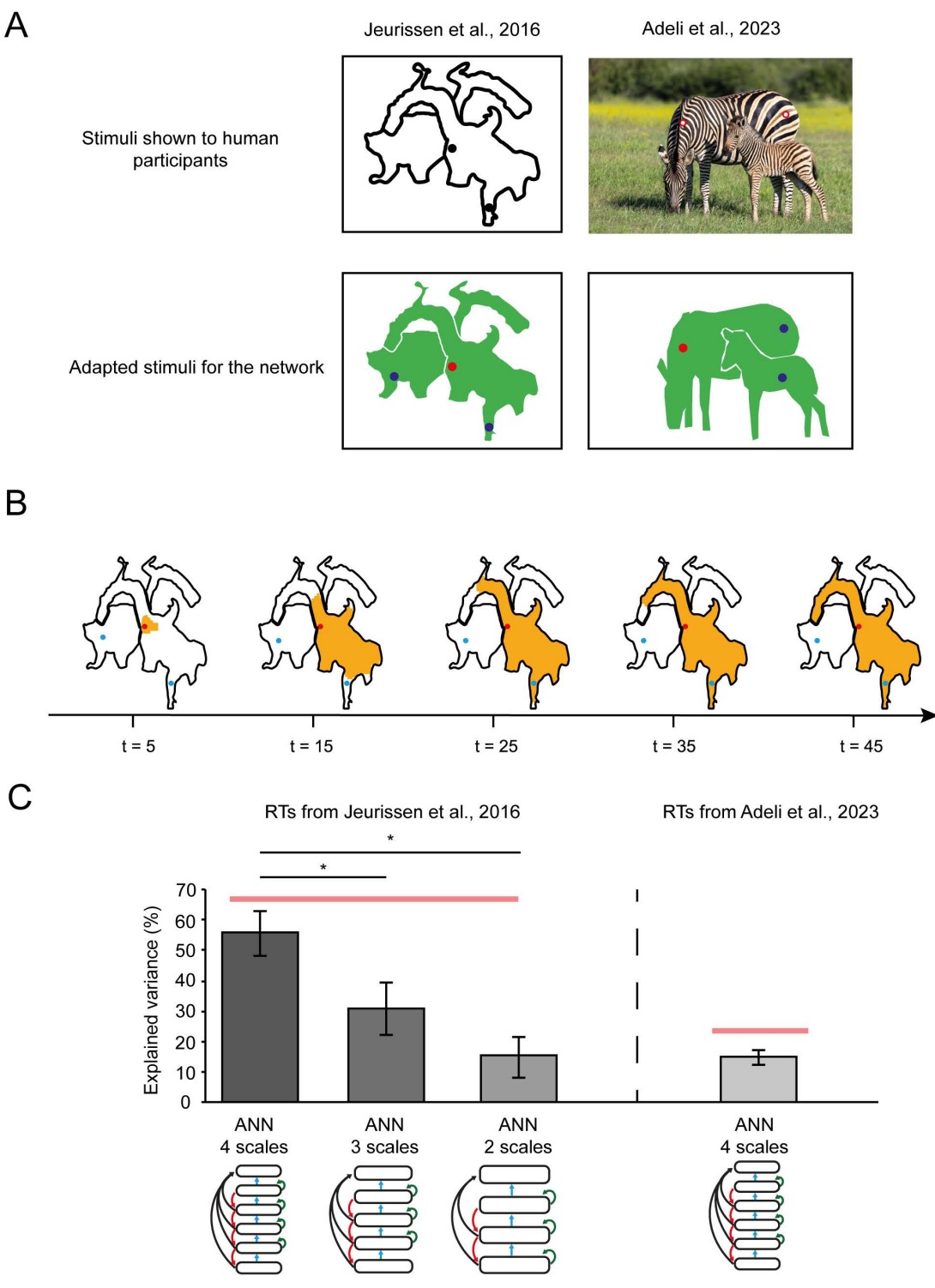

**Fig 4. Object-parsing task. A.** We examined how well the model could explain human RTs in image parsing tasks. We presented simplified images where objects were filled homogeneously and background locations were empty. Image: *Zebra mom and foal* by flowcomm, licensed under Creative Commons Attribution 4.0 International (CC BY 4.0) (https://flic.kr/p/2rRCJVY). **B.** Illustration of the spread of enhanced activity (orange) starting from the red dot. Large homogeneous image regions are filled faster (steps 5-15) than narrower regions (steps 25-35). **C.** Explained variance in human RTs collected by Jeurissen et al. (10) and Adeli et al. (19). We fitted the RTs with network models with either 2, 3 or 4 scales. Note that the quality of the fit improves for more scales. Error bars, standard errors. Red line, noise ceiling. *, P < 0.05.

predictive power approached that of the growth-cone model, which had not yet been implemented as neural network and did not commit to a specific number of scales.

The model thereby goes beyond previous work on incremental grouping by neural networks. It learned to trace curves by trial-and-error at multiple scales, whereas previous networks for curve-tracing used a predetermined connectivity scheme and did not generalize to the parsing of 2D image regions [16,45]. The model also extends studies [46,47] that introduced a specialized hGRU unit for horizontal interactions to solve image parsing tasks. These models computed a measure of uncertainty, which had to be transformed into a measure of human RT and they accounted for less variance than the present model (21% vs. 55% for Jeurissen et al. [15], and 7% vs. 15% for Adeli et al. [43]).

The new model incorporated several key features inspired by the neurophysiology of the visual cortex. Firstly, we created a hierarchically organized network with larger RFs in higher layers, which enabled the faster grouping of straight curves and homogeneous image regions [16]. Secondly, we used a feedforward and a recurrent network with separate roles. Units of the feedforward network provided a veridical representation of the stimulus, providing a stable scaffold for the spreading of enhanced activity in the recurrent network. There are neurons with similar properties in input layers 4 and 6 of the visual cortex of monkeys [22–24], which do not participate in incremental grouping. We trained the feedforward units to only respond to image regions that unambiguously belonged to a single object, enabling the network to select the appropriate scale. The feedforward network gated the recurrent units and thereby enabled a set of recurrent interactions that was appropriate for the scale of the stimulus in the RF [29]. Neurons in the recurrent network learned to spread enhanced activity over the representation of the target object. In the cortex, such neurons that participate in incremental grouping are prominent in layers 2, 3 and 5 [22,24]. Thirdly, we implemented a disinhibitory scheme in which VIP interneurons inhibited SOM interneurons, thereby disinhibiting pyramidal units. Our implementation is consistent with experimental evidence in the visual cortex of mice showing that visual stimuli in the surround of the RF of pyramidal neurons elicit SOM mediated inhibition [48–51]. VIP neurons can release this inhibition by inhibiting SOM neurons, thereby disinhibiting the pyramidal cells [28,51–53]. This interaction between VIP, SOM and pyramidal neurons contributes to figure–ground segregation [18,54], just as in the present model. This disinhibitory circuit has computational advantages, because the maximal activity of the excitatory units in the circuit is bounded by their feedforward input, preventing run-away excitation and guaranteeing that the network reaches a stable state. A variant of such a disinhibitory loop was used in previous work on how networks learn to track moving objects across successive frames of a movie [55]. Here this disinhibitory circuit enabled the network to trace long curves, overcoming the attenuation of the response enhancement occurring in previous work [17].

The present study may inspire future work on the neuronal mechanisms underlying image parsing. One limitation of the present approach is that we pretrained feedforward units to detect stimulus configurations permitting grouping at larger spatial scales, before training the recurrent network to trace curves using reinforcement learning. In the developing visual system, sensitivity to colinear and other grouping cues emerges gradually and learning continues into late childhood [56], suggesting that feedforward representations and grouping dynamic likely co-develop rather than being strictly serial. In a previous study on curve-tracing with only a single scale [17], we were able to train the feedforward and recurrent networks jointly, in an end-to-end manner. The strategy did, however, not consistently elicit grouping across all scales in the present multiscale architecture, although a form of disinhibitory propagation did emerge. This result indicates that learning grouping across multiple scales imposes additional constraints that are not automatically satisfied by our reinforcement learning scheme.

Future studies could investigate whether joint training of feedforward and recurrent pathways can be achieved by introducing appropriate regularization terms, curriculum strategies, or alternative architectural designs. More broadly, other perceptual grouping cues, like similarity of motion, color or luminance [57], are also acquired during development. A compelling direction for future work would be to develop biologically realistic learning rules that allow the simultaneous emergence of such grouping heuristics at higher network levels, together with the propagation of enhanced neuronal activity that underlies incremental grouping.

It would also be of interest to generalize the present approach to segment natural images, which is a process that depends on object-recognition [58]. Indeed, humans parse upright images more efficiently than images that are presented upside down [41,59], illustrating how object recognition aids image parsing. Although the segmentation of natural images is a largely solved problem for transformer-based deep neural networks, the mechanisms for image parsing in the human brain remain only partially understood. The present approach could be generalized to model the recognition of objects in cortical areas, which provide feedback to label individual object parts and low-level features, represented in lower visual cortical areas with enhanced neuronal activity.

In conclusion, the present results provide insight into how brain-like networks learn to integrate grouping cues represented at lower and higher network levels by the spread of enhanced neuronal activity. At a psychological level of description, this process maps onto the spread of object-based attention across all features that are integrated in coherent object representations [10,11]. We look forward to future work, leveraging the highly productive convergence between machine learning, neuroscience and perceptual psychology, to help us better understand how rich, multi-feature object representations emerge in our conscious perception.

## Materials and methods

### Visual stimuli

We probed the capacity of the network to select the blue pixel that was on the same curve or object as the red pixel. We generally used images with 108x108 pixels with three color channels; red, blue and green (Fig 1C). To test scale selection, we also presented images with 144x144 pixels for the curve tracing task, and 594x594 pixels for the object parsing task. The larger images did not require fine-tuning because we used weight-sharing.

### Estimation of human RTs

To analyze the pattern of human RTs, we followed the procedures of the studies where they were gathered. To model the results of Jeurissen et al. [15], we analyzed RTs on correct trials with the fixation point and cue on the same object. We removed outliers that deviated from the mean of the 1/RT distribution by more than 2.5 standard deviations. We averaged the RTs across participants, so that there was one RT for every combination of image (20 images) and cue (3 cue positions per image). For Adeli et al. [21] we also analyzed RTs on correct trials with the fixation point and cue on the same object. We computed the average RTs across participants for every combination of image (255 images) and cue (2 cue positions per image). We performed a regression analysis to test how well RTs were predicted by the growth-cone model, and the artificial neural network.

We computed the standard error for the coefficients of determination ($R^2$) using Cohen et al. (2003) [60] formula (p. 88):

$$SE_{R^2} = \sqrt{\frac{4R^2(1-R^2)^2(n-k-1)^2}{(n^2-1)(n+3)}}$$

Where n is the number of observations and k is the number of independent variables.

To compare $R^2$ values between the model with 4 scales and models with 2 or 3 scales, we computed the standard error of the difference:

$$SE_{diff} = \sqrt{SE_1^2 + SE_2^2}$$

Then, we calculated the Z-score:

$$Z_{diff} = \frac{R_1^2 - R_2^2}{SE_{diff}}$$

Finally, we determined the p-value using:

$$p = 2 \times P(Z > |Z_{diff}|)$$

## Supporting information

**S1 Text. Alternatives for the disinhibitory connection scheme. Fig A. Comparison of the model with disinhibition to models composed of excitatory units and a squashing non-linearity.**
(DOCX)

## Author contributions

**Conceptualization:** Sami Mollard, Sander M. Bohte, Pieter R. Roelfsema.

**Formal analysis:** Sami Mollard.

**Funding acquisition:** Sander M. Bohte, Pieter R. Roelfsema.

**Investigation:** Sami Mollard.

**Software:** Sami Mollard.

**Supervision:** Sander M. Bohte, Pieter R. Roelfsema.

**Writing – original draft:** Sami Mollard.

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
