## [Decision Letter · Decision Letter 0]

24 Dec 2025

PCOMPBIOL-D-25-01220

How the visual brain can learn to parse images using a multiscale, incremental grouping process

PLOS Computational Biology

Dear Dr. Mollard,

Thank you for submitting your manuscript to PLOS Computational Biology.

We thank you for your patience during the review process, which has taken longer than usual.

Your manuscript has now been carefully evaluated by three experts. As reflected in the reviewers’ comments attached below, all three reviewers have expressed positive, and in some cases strongly positive, assessments of your work. In particular, they highly appreciate the significance of the study as well as the clarity and quality of its presentation. It is a 'strong' manuscript.

At the same time, the reviewers have provided a number of constructive comments and suggestions that we believe will help further strengthen the manuscript. We therefore invite you to revise your manuscript by carefully addressing the points raised in the reviews. Subject to satisfactory revision, we are optimistic that the manuscript will be suitable for acceptance in PLOS Computational Biology.

We look forward to receiving your revised manuscript.

Kind regards,

Haojiang Ying, Ph.D.

Academic Editor

PLOS Computational Biology

Andrea E. Martin

Section Editor

PLOS Computational Biology

**Journal Requirements:**

3) Some material included in your submission may be copyrighted. According to PLOSu2019s copyright policy, authors who use figures or other material (e.g., graphics, clipart, maps) from another author or copyright holder must demonstrate or obtain permission to publish this material under the Creative Commons Attribution 4.0 International (CC BY 4.0) License used by PLOS journals. Please closely review the details of PLOSu2019s copyright requirements here: PLOS Licenses and Copyright. If you need to request permissions from a copyright holder, you may use PLOS's Copyright Content Permission form.

Potential Copyright Issues:

- Please confirm (a) that you are the photographer of Figure 4A., or (b) provide written permission from the photographer to publish the photo(s) under our CC BY 4.0 license.

4) Please amend your detailed Financial Disclosure statement. This is published with the article. It must therefore be completed in full sentences and contain the exact wording you wish to be published.

State what role the funders took in the study. If the funders had no role in your study, please state: "The funders had no role in study design, data collection and analysis, decision to publish, or preparation of the manuscript.".

**Reviewers' comments:**

Reviewer's Responses to Questions

**Comments to the Authors:**

Reviewer #1: The authors present an artificial neural network (ANN) implementation of the growth-cone model in order to explain neurophysiological and psychophysical findings. They trained an ANN comprising two separate units and a disinhibitory circuit within the recurrent network, modelling the differences in activity propagation and reaction times for curve-tracing and object-parsing tasks. The advantage of the proposed network is that it is biologically plausible and stable, and in the case of curve tracing, it can be extended to cover long curves. The paper is clear, and the results demonstrate that this architecture can explain previous findings using a biologically plausible neural network. Based on my limited knowledge of incremental grouping, this work appears novel and worthy of acceptance for publication in PLOS Computational Biology.

Minor comments:

L271: 'using backpropagation with a binary cross entropy loss.'

L331: 'measured at the timestep'

L334: 'just as what has been observed'

Figure 3D: the inter model variability is never mentioned, it can be interesting to plot error bars and comment a bit more on whether all models had the same strategy and results (if it was studied)

Figure 4C: authors observe the explained variance evolution when decreasing the number of scales. Is it complicated to increase the number of scales ? (I guess removing some layers of the trained network is easy compare to retraining a bigger model but increasing the amounts of scale and observing an increase in explained variance would provide a stronger proof than an ablation study)

I wonder to what extend the training procedure impacts the model's performance and if a hard-wired architecture for both the FF model (that detects either connected and colinear units or all active units) and the RNN propagating activity only to the neighbors with local connections can also easily account for the results. This question is a bit redundant with the question on how variable models are. Whether if the heuristics used here constraint the model in a way that learning may not be necessary.

A similar disinhibitory circuit has been identified as an actor of contextual modulation (https://www.cell.com/neuron/fulltext/S0896-6273(20)30891-6). It can be interesting to comment on whether the current model aligns with or differs from observed activity modulation in context modulation.

Reviewer #2: This is a strong and easy-to-read manuscript that provides a recurrent neural network implementation of a mechanistic account of the growth-cone model of perceptual grouping. The model performed well, reproducing findings from previous research, and I found particularly compelling its ability to generalize to unseen, more difficult task variants as well as to extend to a separate task, such as object parsing. I have only a few relatively minor comments and suggestions, mostly related to clarity and reproducibility.

Comments

- I was unable to access the GitHub link to examine the data or code repository given in the “Data and Code Availability” statement. I recommend that the authors double-check the repository link and ensure all necessary materials are publicly available.

- It is explained that the neural network includes four spatial scales and qualitatively describes how receptive fields differ across them; however, the exact receptive field dimensions (e.g., kernel sizes, scaling relationship between layers, etc.) are not reported. Additionally, it would strengthen the biological interpretation to comment on whether the scaling of receptive field sizes across layers bears any relationship to RF scaling in the visual cortex.

- Line 193: Extra t? Otherwise, the t is missing from Equation 1.

- Equation 6: Please define the W variable in-text.

- Figure 1, Panel B: The figure caption could be more descriptive. I am assuming the second panel is the difference between the target and distractor responses, but that is not explicitly stated in the figure caption.

- Figure 2, Panel B: The excitatory and inhibitory connections (circles vs. triangles) are difficult to differentiate at their size, especially in Layer l-1. Enlarging the symbols could improve readability.

Reviewer #3: Mollard et al. propose a biologically plausible computational model of segmentation in the early visual cortex that performs well on (and generalizes between) two tasks: curve tracing, and naturalistic object parsing. The model uses a biologically motivated architecture in which recurrent interaction is implemented by disinhibition, rather than direct excitation, which allows the model to propagate activities over longer curves while keeping its activation stable. The model can reproduce several findings from psychophysics and neurophysiology, such as the slowing down of activity propagation in the presence of a nearby distractor curve. The paper is well written and motivated, and the results look solid and a meaningful improvement on the authors’ prior work (e.g. Mollard et al. 2024). There are, however, some issues which I believe the authors should address. In particular, I have a few major comments:

- Several details of the model architecture are unclear, particularly the feedforward component. In Eq. 1, it is clearly stated that the feedforward pass is based on convolutions. The details of these convolutional layers, however, such as the kernel size and stride and the number of channels, are not specified. Also, eq. 1 indicates the feedforward weights as FF_1, with no subscript or superscript corresponding to the layer. Does this mean that a single set of feedforward weights was shared across layers? I assume not, but if that is not the case the notation should be made clearer. I understand that this work is based on Mollard et al. 2024, and I imagine that many of the details are consistent with that paper (e.g. perhaps the 3x3 kernels enforcing a Von Neumann neighborhood), but there are several differences as well. For example, there was no separate training of the feedforward and feedback components in that previous paper. So I believe the authors should make it clearer in which aspects the two feedforward architectures differed and in which aspects they matched. In general, I think important details such as the exact architecture of the models are worth mentioning in this paper, even when they overlap with previous work, to make the paper more self-contained and easily readable. Finally, the authors should release the model’s code - there is a link, but it does not work (https://github.com/samimol/multiscale_tracing). Having access to the code would have disambiguated many of these architectural details.

- Related to the previous comment, also regarding the feedforward network, the way that the feedforward architecture’s task for its separate training was operationalized was not fully clear from the manuscript. My best guess as to how it worked is the following: an NxN grid (corresponding to the dimensions of the downsampled feature map after applying the convolutional layer, so dependent on the spatial scale) containing binary values (1 only if the contour within that location was fully colinear and connected, 0 otherwise) was provided to the network as a target. That was compared with the single layer’s output feature map using cross-entropy, and this error was backpropagated through the single convolutional layer. Is this how it was implemented? I assume that each convolutional layer’s output was a binary map, indicating whether a colinear edge was present in each unit’s RF. The authors should make this clearer, as it is not explicit in the text. Here again, architectural details of the feedforward network do matter, as the supposed “locality” of the feedforward task (which the authors contrast with the spatial interactions happening in the recurrent part of the model) depends on the size of the convolutional kernels. Also, the fact that the different spatial scales were trained independently should be explained more explicitly, as this idea might be counterintuitive to readers (like myself) who might associate convolutional layers and backpropagation with a hierarchical convolutional architecture trained end-to-end, rather than each layer being trained separately on a different spatial scale. Finally, was the separate training of the different spatial scales done with image inputs (i.e. subsampled versions of the same image, like those shown in figure 4D) or feeding each layer’s output to the layer above it? If each layer’s output was a binary map of edge presence, as I guessed above, then it would be possible to train the layer using image inputs and then deploy it on the outputs of the previous layers, but this is not clear from the manuscript.

- The model’s architecture is based on well-motivated biological principles, but it is not clear what the role of some its components is. One of these components is the two-stage training of the feedforward and recurrent networks. The authors mention that this choice was meant to be analogous to the tuning of feedforward neurons during visual experience. However, I believe they should clarify how crucial this separate training was to the results in the paper. As far as I can tell, the model in their 2024 paper was trained jointly using RELEARNN. Since the model in this paper builds upon that previous model (replacing excitation with disinhibition to generate more stable dynamics and allow longer-range enhancement), this mismatch complicates the comparison. Would the network still be able to effectively parse the input images if it were trained jointly? Or is it somehow the case that direct excitation allows the feedforward units’ selectivity to emerge through joint training, but this disinhibition mechanism doesn’t? The authors should clarify this point, if possible by attempting to train the whole system jointly. If that is not feasible, they should clarify why that is the case. Of course, the authors do include a more direct comparison (on p. 24, lines 486-488) with an excitatory network. This comparison is very informative, but I believe the authors should include more details about it, for example by visualizing both the ReLU and squashing models’ performance (e.g. heatmaps of their curve tracing, summary statistics of their performance) side by side with the disinhibitory network.

- Another architectural component the role of which is not clear are the skip connections from each layer to the output layer. I believe the presence of skip connections complicates the interpretation of the feedback connections’ role in mediating interactions across scales. It is possible, for instance, that the need for cross-scale integration is reduced by the fact that the network can combine the outputs of different layers using skip connections. The authors should discuss what the functional role of the skip connections is, and their relation to other architectural components (especially feedback connections) if possible also trying to ablate them.

Besides these major issues, I list some minor issues below.

- A potentially relevant paper to cite in the introduction, where previous work on disinhibition in segmentation and attentional selection is mentioned (lines 206-207), is the following:

Linsley, D., Malik, G., Kim, J., Govindarajan, L. N., Mingolla, E., & Serre, T. (2021). Tracking without re-recognition in humans and machines. *Advances in neural information processing systems*, *34*, 19473-19486.

That paper incorporated a disinhibitory mechanism into a recurrent neural network trained to track objects across time in synthetic and natural videos. I see some analogy between the approach of this manuscript and that paper: where this manuscript describes a selective spread of activity across space, that paper does it across time. However, this is just a suggestion: this is not my field of expertise and there might be several more relevant papers which use similar mechanisms.

- On p. 10, line 193 there is an additional *t* which looks like a typo

- In Figure 2C and in the caption of Figure 4, the term “object-tracing task” is used, while in the text this is referred to as “object-parsing” task. I would encourage the authors to keep the terminology consistent to improve readability.

- Beyond including more details on the feedforward architecture (which I have already mentioned above), the authors should include some more details on its training procedure, such as the number of iterations, and the criterion for stopping.

- On p. 14, lines 281-282, “the activity of input neurons and units of the feedforward network determines the propagation of activity until convergence to a stable state”. I did not understand what this means exactly - isn’t the convergence determined by the activity of the full network, including the recurrent connections? The authors should clarify this.

- On p. 15, lines 313-314, “All networks converged (criterion of 85% correct) within an average of 23,200 trials”. Could the authors include more information (perhaps a supplementary plot) about the eventual accuracy which the networks reached, and how many iterations each of them took to reach it? Especially because the rest of the text seems to imply that the networks were 100% accurate in the task, so it would be useful to clarify this.

- A minor notation inconsistency: in equation 6, the time index T is in boldface, while in the text just before it (and in other equations, e.g. t, t-1) it is not.

- On p. 16, lines 322-323: “Through trial-and-error learning, the network learns that the target curve begins with a red pixel. Specifically, learning increases the feedforward weights from the red pixel start so that the units that represent this location overcome SOM inhibition”. Perhaps this will become clearer when the authors clarify the details of the feedforward architecture that I mentioned previously, but how does this work precisely? Does it mean that the convolutional kernel has higher weights for one of the input channels (corresponding to the color red)?

- P. 18, line 355: “a cue *feel* (I guess the authors meant “fell”) on the same object as the fixation point.”

- P. 18, line 375: in the text a noise ceiling of 67% is mentioned, but in Figure 4C, the red line is above 70%. Could the authors clarify this discrepancy?

- Related to the comparison of the present model with the growth-cone model, shown in Figure 4C, do the authors expect that increasing the number of layers further would make the model reach the growth-cone model’s explained variance, or perhaps even surpass it, and reach the noise ceiling? Would it be possible to test this?

Overall, I believe that the paper is very strong, and I have given the indication “minor revisions” despite the length of my comments. The majority of my comments are requests for clarification, but I understand that some of them might be more labor-intensive (e.g. clarifying the role of the separate feedforward and feedback training, or of the skip connections).

**Have the authors made all data and (if applicable) computational code underlying the findings in their manuscript fully available?**

The PLOS Data policy requires authors to make all data and code underlying the findings described in their manuscript fully available without restriction, with rare exception (please refer to the Data Availability Statement in the manuscript PDF file). The data and code should be provided as part of the manuscript or its supporting information, or deposited to a public repository. For example, in addition to summary statistics, the data points behind means, medians and variance measures should be available. If there are restrictions on publicly sharing data or code —e.g. participant privacy or use of data from a third party—those must be specified.requires authors to make all data and code underlying the findings described in their manuscript fully available without restriction, with rare exception (please refer to the Data Availability Statement in the manuscript PDF file). The data and code should be provided as part of the manuscript or its supporting information, or deposited to a public repository. For example, in addition to summary statistics, the data points behind means, medians and variance measures should be available. If there are restrictions on publicly sharing data or code —e.g. participant privacy or use of data from a third party—those must be specified.requires authors to make all data and code underlying the findings described in their manuscript fully available without restriction, with rare exception (please refer to the Data Availability Statement in the manuscript PDF file). The data and code should be provided as part of the manuscript or its supporting information, or deposited to a public repository. For example, in addition to summary statistics, the data points behind means, medians and variance measures should be available. If there are restrictions on publicly sharing data or code —e.g. participant privacy or use of data from a third party—those must be specified.requires authors to make all data and code underlying the findings described in their manuscript fully available without restriction, with rare exception (please refer to the Data Availability Statement in the manuscript PDF file). The data and code should be provided as part of the manuscript or its supporting information, or deposited to a public repository. For example, in addition to summary statistics, the data points behind means, medians and variance measures should be available. If there are restrictions on publicly sharing data or code —e.g. participant privacy or use of data from a third party—those must be specified.

Reviewer #1: None

Reviewer #2: **No:** The GitHub link provided in the "Data and Code Availability" section does not work/is broken.The GitHub link provided in the "Data and Code Availability" section does not work/is broken.The GitHub link provided in the "Data and Code Availability" section does not work/is broken.The GitHub link provided in the "Data and Code Availability" section does not work/is broken.

Reviewer #3: **No:** As already mentioned in my comments, the link to the repository (https://github.com/samimol/multiscale_tracing) does not work.As already mentioned in my comments, the link to the repository (https://github.com/samimol/multiscale_tracing) does not work.As already mentioned in my comments, the link to the repository (https://github.com/samimol/multiscale_tracing) does not work.As already mentioned in my comments, the link to the repository (https://github.com/samimol/multiscale_tracing) does not work.

PLOS authors have the option to publish the peer review history of their article (what does this mean?). If published, this will include your full peer review and any attached files.). If published, this will include your full peer review and any attached files.). If published, this will include your full peer review and any attached files.). If published, this will include your full peer review and any attached files.

...

Reviewer #1: **Yes:** Antoine GrimaldiAntoine GrimaldiAntoine GrimaldiAntoine Grimaldi

Reviewer #2: No

Reviewer #3: No

**Figure resubmission:**
---

## [Decision Letter · Decision Letter 1]

1 Apr 2026

Dear Mr Mollard,

We are pleased to inform you that your manuscript 'How the visual brain can learn to parse images using a multiscale, incremental grouping process' has been provisionally accepted for publication in PLOS Computational Biology.

Best regards,

Haojiang Ying, Ph.D.

Academic Editor

PLOS Computational Biology

Andrea E. Martin

Section Editor

PLOS Computational Biology

Reviewer's Responses to Questions

**Comments to the Authors:**

Reviewer #3: I thank the authors for thoroughly addressing my comments. I believe they have done so satisfactorily, and that the manuscript has been substantially strengthened and clarified by their additional work. I have no further comments, and I recommend acceptance.

**Have the authors made all data and (if applicable) computational code underlying the findings in their manuscript fully available?**

The PLOS Data policy requires authors to make all data and code underlying the findings described in their manuscript fully available without restriction, with rare exception (please refer to the Data Availability Statement in the manuscript PDF file). The data and code should be provided as part of the manuscript or its supporting information, or deposited to a public repository. For example, in addition to summary statistics, the data points behind means, medians and variance measures should be available. If there are restrictions on publicly sharing data or code —e.g. participant privacy or use of data from a third party—those must be specified.requires authors to make all data and code underlying the findings described in their manuscript fully available without restriction, with rare exception (please refer to the Data Availability Statement in the manuscript PDF file). The data and code should be provided as part of the manuscript or its supporting information, or deposited to a public repository. For example, in addition to summary statistics, the data points behind means, medians and variance measures should be available. If there are restrictions on publicly sharing data or code —e.g. participant privacy or use of data from a third party—those must be specified.requires authors to make all data and code underlying the findings described in their manuscript fully available without restriction, with rare exception (please refer to the Data Availability Statement in the manuscript PDF file). The data and code should be provided as part of the manuscript or its supporting information, or deposited to a public repository. For example, in addition to summary statistics, the data points behind means, medians and variance measures should be available. If there are restrictions on publicly sharing data or code —e.g. participant privacy or use of data from a third party—those must be specified.requires authors to make all data and code underlying the findings described in their manuscript fully available without restriction, with rare exception (please refer to the Data Availability Statement in the manuscript PDF file). The data and code should be provided as part of the manuscript or its supporting information, or deposited to a public repository. For example, in addition to summary statistics, the data points behind means, medians and variance measures should be available. If there are restrictions on publicly sharing data or code —e.g. participant privacy or use of data from a third party—those must be specified.

Reviewer #3: Yes

PLOS authors have the option to publish the peer review history of their article (what does this mean?). If published, this will include your full peer review and any attached files.). If published, this will include your full peer review and any attached files.). If published, this will include your full peer review and any attached files.). If published, this will include your full peer review and any attached files.

...

Reviewer #3: **Yes:** Giacomo AldegheriGiacomo AldegheriGiacomo AldegheriGiacomo Aldegheri

---

## [Editor Report · Acceptance letter]

PCOMPBIOL-D-25-01220R1

How the visual brain can learn to parse images using a multiscale, incremental grouping process

Dear Dr Mollard,

I am pleased to inform you that your manuscript has been formally accepted for publication in PLOS Computational Biology. Your manuscript is now with our production department and you will be notified of the publication date in due course.

With kind regards,

Judit Kozma
